# The Role of Cocrystallization-Mediated Altered Crystallographic Properties on the Tabletability of Rivaroxaban and Malonic Acid

**DOI:** 10.3390/pharmaceutics12060546

**Published:** 2020-06-12

**Authors:** Dnyaneshwar P. Kale, Vibha Puri, Amit Kumar, Navin Kumar, Arvind K. Bansal

**Affiliations:** 1Department of Pharmaceutics, National Institute of Pharmaceutical Education and Research (NIPER), S.A.S. Nagar 160062, India; kale_pep15@niper.ac.in; 2Bristol Myers Squibb, 556 Morris Avenue, Summit, NJ 07901, USA; puri.vibhaa@gmail.com; 3Department of Mechanical Engineering, Indian Institute of Technology (IIT) Ropar, Rupnagar 140001, India; amitkamboj310@gmail.com (A.K.); nkumar@iitrpr.ac.in (N.K.)

**Keywords:** cocrystal, compaction, nanoindentation, slip plane, tabletability, surface topology, interparticulate bonding area, interparticulate bonding strength

## Abstract

The present work aims to understand the crystallographic basis of the mechanical behavior of rivaroxaban-malonic acid cocrystal (RIV-MAL Co) in comparison to its parent constituents, i.e., rivaroxaban (RIV) and malonic acid (MAL). The mechanical behavior was evaluated at the bulk level by performing “out of die” bulk compaction and at the particle level by nanoindentation. The tabletability order for the three solids was MAL < RIV < RIV-MAL Co. MAL demonstrated “lower” tabletability because of its lower plasticity, despite it having reasonably good bonding strength (BS). The absence of a slip plane and “intermediate” BS contributed to this behavior. The “intermediate” tabletability of RIV was primarily attributed to the differential surface topologies of the slip planes. The presence of a primary slip plane (0 1 1) with flat-layered topology can favor the plastic deformation of RIV, whereas the corrugated topology of secondary slip planes (1 0 2) could adversely affect the plasticity. In addition, the higher elastic recovery of RIV crystal also contributed to its tabletability. The significantly “higher” tabletability of RIV-MAL Co among the three molecular solids was the result of its higher plasticity and BS. Flat-layered topology slip across the (0 0 1) plane, the higher degree of intermolecular interactions, and the larger separation between adjacent crystallographic layers contributed to improved mechanical behavior of RIV-MAL Co. Interestingly, a particle level deformation parameter H/E (i.e., ratio of mechanical hardness H to elastic modulus E) was found to inversely correlate with a bulk level deformation parameter σ_0_ (i.e., tensile strength at zero porosity). The present study highlighted the role of cocrystal crystallographic properties in improving the tabletability of materials.

## 1. Introduction

Oral solid dosage forms (such as tablets and capsules) represent the most popular delivery system amongst various pharmaceutical dosage forms, as they offer many benefits in terms of cost, stability, ease of handling, and patient compliance [1]. The mechanical properties of active pharmaceutical ingredients (APIs) play a critical role in the manufacturing of solid dosage form, as poor mechanical properties can cause difficulties during processing. With the emergence of crystal engineering, cocrystallization has become a useful strategy in the design of pharmaceutical materials with desired properties [2,3,4,5]. Cocrystallization has received significant attention because it can be employed for altering physicochemical properties of the APIs such as solubility [6], dissolution [7,8], stability [9], hygroscopicity [10,11], and mechanical properties [12,13]. Many reports suggest that cocrystallization can improve [14,15], deteriorate [16,17] or have no impact on modulation of mechanical properties of organic molecular solids including APIs. Understanding the role of crystallographic/supramolecular features, in modulating mechanical properties, remains an area of significant interest.

Tabletability is represented by tensile strength as a function of applied compaction pressure. It has been commonly used to compare the compaction performance of organic molecular solids and is particularly useful in studying bulk deformation behavior [18,19]. The tensile strength of the compacts is governed by interparticulate bonding area (BA) as well as interparticulate bonding strength (BS) [20]. Therefore, a thorough understanding of tabletability can be developed by a quantitative model, based on the concept of interparticulate bonding area and bonding strength (BABS model) [21,22]. Increased BA is achieved on irreversible (plastic) deformation of organic molecular solid and this deformation behavior is known as plasticity [21]. Crystallographic features such as slip systems are responsible for plasticity (increase in BA) in single component polymorphs [19,23] and multicomponent cocrystals [14,24]. The degree of intermolecular interactions and true density contribute to the BS of the crystal form [19,20,25,26]. It is worthwhile mentioning that “plasticity” only contributes to BA, and hence the role of BS also needs to be considered while evaluating bulk deformation (compaction) behavior.

Nano-mechanical parameters such as elastic modulus (E), indentation hardness (H) and elastic recovery (determined from load-displacement (*P–h*) curves) provide useful insights on the mechanical behavior of solids at a particle (crystal) level [27,28,29]. It is noteworthy that nanoindentation focuses on particle mechanical properties, but other factors such as particle size, particle shape and distribution, inter-particle interaction can have a significant effect when translating these properties to the bulk deformation (compaction). Interestingly, nanoindentation is useful for interpolation of particle-level behavior to crystallographic features [30], thereby furthering our understanding of crystal structure‒mechanical property relationships. As demonstrated by Wendy and Duncan-Hewitt [31], and recently by our research group [32], the indentation hardness to elastic modulus (H/E) ratio is a predictor of compaction behaviour of materials undergoing deformation. The higher the value of H/E, the higher are the residual stresses after compaction (higher elastic recovery) and vice-versa. Another predictor of elastic recovery is (*h_max_*‒*h_p_*), wherein *h_p_* is the permanent indentation depth after removal of the test load and *h_max_* is indentation depth at a maximum load; both are computed from the (*P–h*) curve.

Crystal anisotropy is responsible for distinct variations of the plastic deformation within the same structural class. Consequently, deformation processes in molecular solids cannot be labelled in simple terms and each molecular solid has to be evaluated individually [29,33]. It is well accepted that the presence of active slip planes in molecular solids leads to “plastic” deformation. A higher plasticity confers improved tabletability, because of increased BA. Therefore, slip planes have been increasingly used to describe “plasticity” and have also been used to correlate to the deformation behavior of molecular solids. However, the existing literature lacks descriptions of how a molecular solid undergoes deformation when its crystal structure contains more than one active slip system with two different surface topologies (flat and zigzag topologies).

In the present work, we reveal the role of crystallographic features such as slip planes, slip plane topology, intermolecular interactions (nature of hydrogen bonding: 1D, 2D, or 3D) and largest d-spacing on “plasticity” and interparticulate bonding strength of the model systems—rivaroxaban (RIV), malonic acid (MAL) and 2:1 rivaroxaban-malonic acid cocrystal (RIV-MAL Co). In particular, we focus on how the materials deform when their crystal structures contain a single active slip plane system (RIV-MAL Co), more than one slip plane system (RIV) and devoid of active slip planes (MAL). In this study, the nanoindentation technique was employed to uncover “crystal level” deformation behavior, while “bulk level” deformation was studied using Compressibility-Tabletability-Compactibility (CTC) profiling. Slip plane systems were predicted by visualization as well as attachment energy calculations, and further experimentally confirmed by nanoindentation studies.

## 2. Materials

Molecular structures of RIV and MAL are presented in Figure 1. RIV (Purity > 99.5%) was generously provided as a gift sample by MSN Laboratories, India. Malonic acid (MAL) and 2,2,2-trifluoroethanol were purchased from Sigma-Aldrich, St. Louis, MO, USA and Spectrochem Pvt. Ltd., Mumbai, India, respectively, and used as received. Crystallization experiments were carried out to generate samples for compaction. For nanoindentation experiments, relatively larger-sized crystals were generated, and the crystallization methodology for same has been thoroughly discussed in our recent publication [10].

## 3. Methods

### 3.1. Crystallization Experiments

#### 3.1.1. RIV for Compaction Study

A saturated solution of RIV was prepared by dissolving the drug (~6.0 g) in acetonitrile (600 mL) with magnetic stirring the solution on a heated water bath at 60‒65 °C for 30 min followed by cooling at room temperature (RT). The resultant solution was kept for slow solvent evaporation at room temperature. Columnar-shaped, small-sized crystals, generated within 1‒2 h, were collected by vacuum filtration and subjected to air drying.

#### 3.1.2. MAL for Compaction Study

Crystalline MAL from the supplier was ground using a mortar and pestle to obtain the desired particle size range of 5‒40 µm. Prior to the compaction study, the powder was dried in a hot air oven at 50 ± 2 °C for 2 h to remove any adsorbed moisture from the surface.

#### 3.1.3. RIV-MAL Co for Compaction Study

The compaction study requires a large quantity of material; hence the previously reported process was scaled up. Initially, a small batch (~150 mg) of RIV-MAL Co was prepared as per the reported method [10]. These crystals were used as “seeds” for preparing a batch of ~8.0 g.

For preparing 8.0 g batch of RIV-MAL Co, 6.0 g of RIV was dissolved in 30.1 mL of 2,2,2-trifluoroethanol (TFE) in a glass beaker (50 mL capacity) with stirring and heated using a water bath between 70 and 80 °C. MAL (2.07 g) was added to the hot solution of RIV and volume of the solution was reduced to approximately 55% by heating (70‒75 °C) it in the open vessel for 1‒2 h. Thereafter, the hot solution was transferred to another glass beaker. The resultant solution was seeded with the crystals of RIV-MAL Co and cooled to room temperature. After 10‒20 min, the cocrystals crystallized from the solution were collected using vacuum filtration. The crystals were dried at 25 °C in a vacuum drier.

All three solids were stored in a desiccator at 43% RH (maintained using a saturated solution of potassium carbonate, K_2_CO_3_) until further use.

### 3.2. Differential Scanning Calorimetry (DSC)

The melting point and heat of fusion of samples were determined using DSC instrument (model Q2000, TA Instruments, New Castle, DE, USA) equipped with a refrigerated cooling system. The instrument was operating with TA Universal Analysis^®^ software (version 4.5 A) and calibrated initially for temperature and heat flow using high purity standard of indium. The sample (2−4 mg) was analyzed in a T_zero_ aluminium pan crimped with a lid. The weight of sample was accurately recorded on a microbalance (Sartorius GM 1202, Goettingen, Germany). Each sample was subjected to thermal scanning at a heating rate of 10.0 °C/min. The purge of dry nitrogen at flow rate of 50 mL/min was maintained during analysis.

### 3.3. X-Ray Powder Diffractometry (XRPD)

XRPD of the samples was collected at room temperature (25.0 ± 2.0 °C) on a Bruker D8 Advance Diffractometer (Bruker, AXS, Karlsruhe, Germany). The instrument was operating with Cu Kα radiation (1.54 Å) passing through a nickel filter, and the tube voltage was set at 40.0 kV and current of the generator (amperage) at 40.0 mA. The sample (250‒300 mg) was mounted on a poly(methyl methacrylate) (PMMA) sample holder and pressed with a glass slide to achieve coplanarity with the surface of sample holder. Each sample was then scanned in 2θ range of 5.0°‒40.0° with a step size of 0.05° and a step time of 1.0 s. The obtained data were plotted using OriginPro software (OriginLab Corporation, Northampton, MA, USA).

### 3.4. Particle Size Distribution (PSD)

Comparable particle sizes were obtained by passing the samples through a British sieve size (BSS) No. 100# and retaining on a 120# sieve. Each sample was mounted on a glass slide with silicone oil and the observed under optical microscope (Leica DMLP, Leica Microsystems, Wetzlar, Germany) for measuring particle diameter. The diameter (i.e., length along the longest axis of individual particles) of 300 particles were measured. Particle size distribution curve was plotted to determine the diameters corresponding to 10%, 50%, and 90% of cumulative undersize particles, i.e., D_10_, D_50_, and D_90_.

### 3.5. Specific Surface Area (SSA) Measurement

The specific surface area of RIV, MAL and RIV-MAL Co was determined by nitrogen gas adsorption using “Smart Sorb 92” surface area analyzer (Smart Instruments, Mumbai, India). The gas loop of the instrument was filled with known amount of sample (500–800 mg) and then submerged into liquid nitrogen for adsorption. After completion of the adsorption cycle, a desorption was triggered by submerging the glass loop into water at ambient condition. The quantity of the adsorbed gas was measured using a thermal conductivity detector and the data were integrated using an electronic circuit in terms of counts. The measured parameters were then used to calculate the surface area of the sample by employing the adsorption theories of Brunauer, Emmett, and Teller. SSA was reported as average of three individual measurements (n = 3).

### 3.6. Moisture Content (MC)

Karl Fischer titration using “Titrino 794 Basic titrator” (Metrohm AG, Herisau, Switzerland) was used to measure moisture content (water content) of samples. Prior to analysis, calibration of the instrument was performed using disodium tartrate dehydrate. MC was reported as the average of three measurement for each sample.

### 3.7. True Density Determination

The true density of samples was determined using a helium gas pycnometer (Pycno 30, Smart Instruments, Mumbai, India) as per the previously reported protocol [34]. Before analysis, the samples were pre-dried at 40 °C for 24 h in a vacuum oven to avoid the effect of residual moisture on true density measurement. The pre-dried sample, sufficient to fill 3/4 of the volume of the sample cell, was weighed (1.5–2.5 g) and transferred into the sample cell. The first pressure reading (P_1_) was recorded after passing a pressurized pure helium gas in a known reference volume into the reference cell. Then, the pressurized helium gas was allowed to flow from the reference cell into the sample cell. This led to drop in the initial pressure that recorded as second pressure reading (P_2_). These values of P1 and P2 put into Equation (1) to calculate the true volume V_p_.
(1)$$V_{P}=\left(\;\frac{P_{1}}{P_{2\;}}-1\right){\left(V_{c}-{\;V}_{r}\right)}_{}$$
where V_c_ and V_r_ are the cell volume and reference volume having values of 18.9522 and 11.9587 cm^3^/g, respectively. True density was calculated by dividing the sample mass by true volume (V_P_) value_._

### 3.8. Preparation of Compacts for Studying Bulk Deformation Behavior

Compacts were prepared by compressing 400 mg of crystalline powder using different compaction pressure in a hydraulic press (Type KP, Sr. No. 1125, Kimaya Engineers, Maharastra, India). The applied dwell time for compaction preparation was 1.0 min using a 13.0 mm punch die set (round, flat-faced punch). Different compression forces were applied manually to achieve a range of compaction pressures from 37‒222 MPa. The actual compaction pressure was determined from the know value of the applied hydraulic load (Force) and the surface area of the flat punch-die set used for compression. Equation (2) was used for converting the applied load into compaction pressures.
(2)$$P=\frac{F}{A\;}$$
where, F is the applied hydraulic load (Newton, N), and A is the surface area of the flat punch-die set (in mm^2^). Prior to analysis, the prepared compacts were stored for 48 h under ambient conditions to allow for relaxation of any residual stress. Subsequently, compacts were analyzed for weight, thickness, and breaking force measurement.

### 3.9. Calculation of Tensile Strength and Porosity

The assessment of tensile strength (σ) value in bulk deformation profiling helps to eliminate the undesirable effect of variable tablet thickness on a measured breaking force. Therefore, tensile strength (σ) was calculated using Equation (3), based on breaking force (F), table diameter (d) and tablet thickness (t).
(3)$$\sigma =\frac{2F}{\pi \mathrm{dt}}$$
where σ is the tensile strength in MPa, F is the observed breaking force in N, d is the diameter in mm, and t is thickness of the compact in mm.

Tablet hardness tester (Erweka, TBH 20, USA) was used to measure the breaking force (F) of all of the compacts. Digital caliper (CD-6 CS, Digimatic Mitutoyo Corporation, Kanagawa, Japan) was used to measure tablet diameter and thickness. The porosity (ε) of the compacts was calculated using Equation (4), from tablet density (ρ_c_) and true density of the powder (ρ_t_). ρ_c_ is calculated from the weight and volume of the resulting tablet, while ρ_t_ is measured by helium pycnometer as described above.
(4)$$\varepsilon =1-\frac{\rho_{c}}{\rho_{t}}$$

### 3.10. Nanoindentation Experimentation

Nanoindentation was performed on oriented single crystals of RIV, MAL and RIV-MAL Co using a Ti-950 TriboIndenter (Hysitron Inc., Minneapolis, MN, USA) using a protocol, which is similar to that of previously reported by our research group [32]. Briefly, the indenter employed was a tripyramidal (Berkovich) tip having an inclined angle of 142.3° and a tip radius of ∼150 nm. The fused silica and polycarbonate standards were used to calibrate the tip area function. The tip area function calibration was carried out by performing a series of indents with different contact depths on a standard sample of known elastic modulus (E). A plot of the calculated area against contact depth (h) was created and fitted by the TriboScan software. An optical microscope integrated into the nanoindentation system was used to identify the regions on crystal surface for testing. The “tip to optics calibration” was undertaken by performing 10 indents in an “H-pattern”. The testing was carried out at 28 ± 0.5 °C temperature and 45 ± 5% relative humidity. For quasi-static analysis of all samples, 10−12 subsequent indents were performed along the length, midline parallel to the longest axis of the crystal on a dominant face with user-specified parameters. The sufficient contact depths, large enough to local surface roughness were estimated to avoid strong effect of roughness on the measured mechanical properties. The peak load (P) for these indentations was 1000 μN, and the indent spacing was 55.0 μm. A load function consisting of a 5 s loading to peak force (F) segment, followed by a 2-s hold segment and a 5-s unloading segment was used (the loading and unloading rates were 0.2 mN/s). The Oliver and Pharr method [35] was employed to compute the nanomechanical hardness (H) and the elastic modulus (*E_r_*). The *E_r_* value is related to the Young modulus of elasticity of the tested sample (*E_s_*) and the indenter (*E_i_*) through the following relationship in Equation (5):(5)$$\frac{1}{E_{r}}=\frac{\left(1-v_{i}^{2}\right)}{E_{i}}+\frac{\left(1-v_{s}^{2}\right)}{E_{s}}\;$$
where *υ_i_* is Poisson ratio for the indenter material, while *υ_s_* is Poisson ratio of the substrate material. The values of the elastic modulus and Poisson ratio for the diamond indenter tip are 1140 GPa and 0.07, respectively.

### 3.11. Molecular Modeling

The crystal structures of MAL, RIV and RIV-MAL Co were examined for identification of slip planes, d-spacing, intermolecular interactions and H-bonding dimensionalities using Mercury software (Version 3.7, Cambridge Crystallographic Data Centre, CCDC, Cambridge, UK). Previously, our lab has successfully solved and deposited crystal structures of RIV and RIV-MAL Co with the CCDC numbers 1854617 and 1854618, respectively [10]. The CIF file of MAL (CSD Reference code MALNAC02, deposition number 1209218) was downloaded from the CCDC website, https://www.ccdc.cam.ac.uk/structures/.

### 3.12. Attachment Energy Calculations

The attachment energy, *E_att_*, is defined as the energy released on the attachment of a growth slice with the thickness *d*_hkl_ to a growing crystal face [36,37]. *E_att_* is calculated as *E_att_ = E_lattice_ − E_slice_*, where *E_lattice_* is the lattice energy of the crystal, and *E_slice_* is the energy released on the formation of a growth slice of a thickness equal to the interplanar d-spacing, *d*_hkl._

Crystal morphology predictions were performed using Material Studio 2018 (Biovia, San Diego, CA, USA). The growth morphology method (with COMPASS II force field) was used to generate *E_att_* of different crystal faces. All calculations were performed at fine quality with “Ewald” electrostatic summation method and “atom-based” van der Waals summation method. The crystallographic planes with least total *E_att_* were identified as slip planes. Additionally, *E_att_* calculations were also performed using Dreiding force field (with “charge Qeq” and “current charge”).

### 3.13. Statistical Analysis

The statistical significance of various bulk deformation parameters was compared using the paired t-test, assuming equal variances using GraphPad Prism 5 software, version 5.04 (GraphPad Software, Inc., San Diego, CA, USA). The test was considered to be statistically significant if *p* < 0.05.

## 4. Results and Discussion

### 4.1. Solid-State Characterization

The overlay of DSC heating scans and XRPD patterns for RIV, MAL and RIV-MAL Co are presented in Figure 2 and Figure 3, respectively. The DSC heating curve of the RIV crystals displayed a melting endotherm at 230.5 °C (T_m_, onset) with a heat of fusion (∆H_f_) of 118.6 J/g, which matches to the melting temperature of rivaroxaban form I (Figure 2). The XRPD pattern of the generated RIV sample exhibited sharp diffraction peaks at 2Ɵ values of 9.1°, 12.2°, 14.4°, 16.6°, 17.6°, 18.2°, 19.6°, 20.0°, 21.8°, 22.6°, 23.5°, 24.1°, 24.8°, 25.8°, 26.8°, 29.6°, 30.3°, 31.9° ± 0.2° (Figure 3). The obtained diffraction pattern matches to the calculated powder pattern (CCDC identification code—LEMSOO01, deposition no. 1854617) [10] and the peak positions correspond to the values reported for RIV form I. The DSC and XRPD data together confirmed the identity and solid form purity of RIV sample generated for the compaction study.

The DSC heating curve of MAL sample showed a broad endothermic event between 85.0° to 109.0 °C (related to solid-solid phase transition) followed by a sharp melting endotherm at 135.0 °C (T_m_, onset), (∆H_f_ ≈ 235.9 J/g) [38]. The XRPD pattern of the sample showed sharp diffraction peaks, which correspond to the β form of MAL.

As shown in the DSC analysis of the RIV-MAL Co, the sample showed a transition point between 110 to 122 °C (related to solid to solid phase transition), followed by the endothermic peak corresponding to cocrystal melting at 167.9 °C (ΔH_f_ is ~36 J/g)[10]. Following the cocrystal melting, the exothermic event (recrystallization of RIV) and the endothermic at ~230 °C (melting of the recrystallized RIV) were observed. The above-thermal events were consistent with the previously reported behaviour of RIV-MAL Co [10]. The experimental diffraction pattern of RIV-MAL Co sample (Figure 3) matched the calculated powder pattern obtained from single-crystal data reported by our laboratory (CCDC identification code- YORVEJ, deposition No. 1854618) [10]. Thus, DSC and XRPD together confirmed a solid form purity of the scale-up batches.

### 4.2. Particle Level and Bulk Level Attributes

Particle size analysis revealed that all three solids had similar *D_50_* and *D_90_* values (Table 1). Similar particle size distribution allows better comparison of compaction behavior of these solids at the molecular and supramolecular level. MAL, RIV, and the cocrystal had a moisture content of less than 0.3% *w/w* (Table 1). MAL had a significantly higher true density value (1.628 ± 0.001 g/cm^3^) as compared to RIV and MAL (1.536 ± 0.005 g/cm^3^ for RIV and 1.534 ± 0.007 g/cm^3^).

### 4.3. Bulk Deformation Behaviour

The deformation behavior of pharmaceutical materials at a bulk level is commonly studied by performing compressibility, tabletability, compactibility (CTC) profiling, as it provides a comprehensive understanding on the role of interparticulate bonding area (BA) and bonding strength (BS) [25,26,39,40,41]. A measure of the ability of powder material to undergo reduction in volume under the application of compaction pressure is known as “compressibility”. Compressibility is represented as a plot of porosity against compaction pressure and the plot signifies the extent of increase in interparticulate bonding area (BA).

The compressibility profiles of the three solids demonstrated greater compressibility of RIV and RIV-MAL Co over MAL at a given compaction pressure (Figure 4a). Slightly higher compressibility of the cocrystal over RIV was observed at higher compaction pressure (>150 MPa) and the differences were statistically significant (*p* < 0.05). The applied compaction pressure may lead to pressure-induced phase transformation in the samples. When examined by using DSC analysis, the compacts of the three molecular solids did not show any evidence of pressure-induced phase transformation.

Compactibility is defined as “*the ability of the powder material to produce tablets of sufficient tensile strength under the effect of densification*” [21,25]. It is represented by a plot of tensile strength against tablet porosity, and the plot signifies the bonding strength (BS) of a given material. Compactibility plots (Figure 4b) indicate a higher bonding strength of the cocrystal (RIV-MAL Co) compared with that of RIV and MAL at all compaction pressures. As shown in the compactibility plot, MAL compacts exhibited reasonably good BS at comparatively larger porosity values as compared to the porosity values of RIV and RIV-MAL Co. The inability of MAL to undergo volume reduction (i.e., lower compressibility) may be a cause of its larger porosity at comparable compaction pressures.

Tabletability is defined as “the capacity of the powder material to be transformed into a tablet of specified tensile strength under the effect of applied compaction pressure” [21,42]. The overall tabletability of a material is governed by both BA and BS. The tensile strength (σ) of all the solids increased with increasing compaction pressure from 37 to 222 MPa (Figure 4c). The tensile strength value of the compacts at higher compaction pressure (222 MPa) was comparatively lowest for MAL (σ = 1.9 MPa), intermediate (slightly higher) for RIV (σ = 2.1 MPa), and the highest for RIV-MAL Co (σ = 3.2 MPa). The tabletability order based on the tensile strength at the higher compaction pressure follows the order MAL < RIV < RIV-MAL Co. The required tensile strength for a tablet to withstand the stresses encountered during its handling and transport is 2.0 MPa, and it should be attained by compressing a material at compaction pressure ≤ 200 MPa. Interestingly, the tabletability plot for RIV-MAL Co indicated that the cocrystal could produce a tablet of sufficient tensile strength (2.3 MPa) at a relatively lower compaction pressure of 37 MPa (please refer Figure 4c). The RIV compacts demonstrated a tensile strength of 2.1 MPa at a compaction pressure of 222 MPa, while the desired tensile strength of 2 MPa could not be achieved for MAL even at a compaction pressure as high as 222 MPa. Hence, the cocrystallization led to significant improvement in the mechanical properties, despite the fact that both parent components (RIV and MAL) possessed relatively poor mechanical properties. Hence, it is interesting to unfold the supramolecular and crystallographic basis of the compaction behavior of these three solids.

#### 4.3.1. Heckel Analysis

The Heckel model provides method for transforming force and displacement signals to a more simpler linear relationship for materials undergoing the compaction process [43]. This increases the popularity of model for studying relationship between relative density and applied compaction pressure. [19,26,44]. The basis for the Heckel equation is that the densification of the bulk powder under applied compaction pressure follows first-order kinetics [44].

The “in-die” measurements do not account for elastic recovery of material, eventually affecting data interpretation and accuracy. Therefore, in the present study, densification of RIV, MAL, and RIV-MAL Co as a function of applied compaction pressure was evaluated by Heckel analysis using “out-of-die” measurement. The “out-of-die” measurement is more reliable method to study bulk deformation of a pharmaceutical material because the material is allowed to undergo elastic deformation prior to density analysis.

The linear region of the Heckel plot provides an important property of the material, that is mean yield pressure (Py). In the Heckel plots, correlation coefficients of R^2^ > 0.98 were achieved in the case of RIV and RIV-MAL Co, while R^2^ > 0.91 was achieved in the case of MAL. Among these solids, RIV-MAL Co showed the highest densification and the lowest Py (Figure 5). The Py of MAL (323 MPa) is very high as compared with RIV (133 MPa) and RIV-MAL Co (83 MPa). The lower mean yield pressure (Py) value obtained for RIV-MAL Co indicates its excellent plastic deformation under the applied compaction pressure. In the same way, RIV can also undergo plastic deformation at the normally utilized compaction pressure of 133 MPa. In contrast, MAL could not undergo plastic deformation at a compaction pressure as high as 323 MPa and also exhibited severe chipping and laminations at compaction pressure > 222 MPa. This behavior demonstrates poor plasticity of MAL during bulk deformation.

#### 4.3.2. Ryshkewitch‒Duckworth Analysis

The compactibility of the material is described by Ryshkewitch‒Duckworth equation (Equation (6) [45]. Tensile strength at zero porosity (σ_0_) can be determined using this mathematical model. Since the tensile strength of a material is normalized by the bonding area at zero porosity; σ_0_ represents interparticulate bonding strength (BS) of the material undergoing compaction. The effect of bonding area was minimal as the similar particle shape and PSD were used for CTC profiling.
(6)$$\sigma =\sigma_{0}\;e^{-a\varepsilon }$$
where σ is tensile strength, α is a constant and ɛ is porosity. In its logarithmic form, a linear relationship between porosity and the log of the tensile strength was obtained. The value of σ_0_ was 2.75, 2.58, and 3.21 MPa for MAL, RIV, and RIV-MAL Co, respectively. The higher σ_0_ value indicates a greater BS for the cocrystal over the API and coformer. At the same time, the BS of MAL was higher than that of RIV. Thus, Ryshkewitch analysis supports the findings of the compactibility plot and confirmed the greater BS of the cocrystal.

The higher tabletability of RIV-MAL Co was the outcome of both higher BA and BS, as demonstrated by its low yield pressure (Py 83 MPa) and high σ_0_ (3.21 MPa) (Figure 5). Comparable tabletability profiles were observed for RIV and MAL, despite the low plasticity of MAL. The tabletability of RIV was predominantly contributed by the BA, as indicated by the low yield pressure (Py of 133 MPa). BS predominantly governs the tabletability of MAL, as supported by the observed σ_0_. The high Py of MAL can be correlated to its brittle nature. Based on the compactibility plots and relative tensile strength at zero porosity (σ_0_), the order of BS in the three solids can be ranked as RIV < MAL < RIV-MAL Co. The Ryshkewitch–Duckworth analysis confirmed the higher BS for MAL (σ_0_ = 2.75 MPa) than RIV (σ_0_ = 2.58 MPa), while the highest BS was observed for RIV-MAL Co (σ_0 =_ 3.21 MPa). In other words, the higher work of adhesion was observed at significantly lower compaction pressure in RIV-MAL Co as compared to RIV and MAL.

### 4.4. Particle Level Deformation: Quantifying Crystal Deformation by Nanoindentation

Crystals with smooth surfaces were subjected to nanoindentation to decipher the particle-level deformation behaviour. The direction of the applied stress was perpendicular to the slip plane predicted based on the visualization and attachment energy calculations (applicable for RIV and RIV-MAL Co). Nanoindentation parameters, i.e., elastic modulus (E), mechanical hardness H, and 1/E values for crystal samples, are presented in Table 2.

Indentation hardness (mechanical hardness) H denotes the resistance offered by the material to plastic deformation [29]. A low value of H is indicative of lower resistance offered by the material to undergo irreversible (plastic) deformation. However, organic molecular solids show initial elastic deformation followed by plastic deformation. Before plastic deformation takes place, the elastic limit has to be exceeded by the applied stress.

E is a measure of the resistance to elastic deformation and is a function of the energy of the interaction between molecules and their distances of separation [29]. The 1/E (compliance) can be correlated with elastic recovery (ER). A high ER indicates dominance of elastic deformation, which adversely contributes to plasticity (BA) and thus to the tabletability of the material.[46]

The lower the value of E, the larger is the 1/E and hence higher will be the elastic recovery. Amongst the three molecular solids, RIV possesses the lowest E (03.41 GPa), indicating the highest elastic recovery. The high elastic recovery of RIV crystals was also verified by the higher (*h_max_**‒**h_p_*) value (178 GPa) (Figure 6), which is the elastic recovery determined from the (*p**‒**h*) loading–unloading curve. The value of (*h_max_**‒**h_p_*) was significantly higher for RIV (178 GPa) as compared to MAL (74 GPa) and RIV-MAL (75 GPa). The high E value (i.e., low 1/E) and low (*h_max_**‒**h_p_*) for MAL and RIV-MAL Co were also indicative of lower ER of the coformer and cocrystal as compared to the API.

The high H of MAL (0.71 GPa) depicts a higher intermolecular interaction between MAL molecules, which can be directly correlated to the crystallographic features of MAL. Careful evaluation of the crystal structure of MAL showed hydrogen bonding interactions along all three axes (3D H-bonding), which provided the greater hardness to MAL (this is thoroughly discussed in Section 4.5.3). The comparatively higher hardness may hinder plastic deformation of MAL crystals when subjected to bulk compaction. RIV-MAL Co showed an “intermediate” value of H when compared with the H values of MAL and RIV (Table 2). At the same time, the cocrystal exhibited low elastic recovery as evidenced by the low values of 1/E and (*h_max_**‒**h_p_*) (Figure 6). Thus, the crystals of cocrystal possess a dominance of plasticity over elasticity.

Wendy and Hewitt in 1989 found that the H/E ratio can be used to predict bulk deformation (compaction) behaviour of materials based on particle-level deformation study using the microindentation technique [31]. In this study, acetaminophen possessed the largest ratio of H/E and exhibited the poorest compaction, i.e., tablets capped and delaminated extensively during decompression and ejection from the die. Adipic acid compacts with a relatively large H/E ratio also underwent delamination during wear testing. Conversely, the materials with a lower H/E ratio could form tablets free from the above defects. This means Hewitt”s work only “qualitatively” correlated H/E ratio with compaction behaviour because the compaction behaviour of the materials was qualitatively described as “good” (for materials whose compacts were free from of capping and lamination) or “poor” (for materials whose compacts showed capping or lamination). Interestingly, the present work provided a correlation of H/E ratio with a “quantitative” bulk deformation parameter, i.e., interparticulate bonding strength at zero porosity (σ_0_). In this work, the H/E was found to inversely correlate to σ_0_ (Figure 6b). The cocrystal had the lowest H/E (0.029) and exhibited the highest σ_0_ (3.21 MPa), while RIV had the highest H/E (0.58) and showed the lowest σ_0_ (2.58 MPa). The value of H/E ratio was “intermediate” for MAL (0.040) and hence MAL exhibited the “intermediate” σ_0_ (2.75 MPa) among the three molecular solids.

### 4.5. Identification of Crystallographic Features

Crystallographic features such as slip planes, topology of slip planes, crystallographic density, molecular packing and nature of intermolecular interactions predominantly govern the deformation behavior at both crystal as well as bulk level. The crystallographic factors influencing the bulk deformation behavior can be divided into two categories—(1) those contributing to increasing BA (plasticity) include—slip planes, topology and numbers of slip planes; (2) those contributing to BS include—the strength of interactions along the weakest crystallographic planes and true density or crystallographic true density. It is well reported that the presence of active slip planes is responsible for plastic deformation (increasing BA) of organic molecular solids including pharmaceutical solids [13,47,48]. Slip planes are defined as “*crystallographic planes in the crystal structure which contain the weakest interaction between the adjacent planes and are accounted by the highest molecular density and largest d-spacing, as compared to the other planes in that crystal*” [23,47,49].

The slip plane identification based on *E_att_* calculation assumes that a plane with the least absolute attachment energy would have the weakest interaction between the adjacent planes and could slip (glide) over one another more easily than other planes in the crystal. The crystal morphology predictions by the “growth morphology” method assumes that the planes with lower attachment energies will grow at a slower rate and hence will be manifested as the larger faces of the crystal habit and vice versa. Typically, the slip plane in a crystal structure is likely to be the one with the least *E_att_* and may manifest as the “largest” face within the crystal habit. The largest surface face (facet) has the highest surface area contribution to the crystal habit, and hence can be experimentally identified by either simple microscopic analysis or more accurately by face indexation analysis. Further, nanoindentation of the crystal provides experimental evidence of the presence or absence of slip planes when the stress is applied normal to the most predominant facet of a crystal (i.e., probable slip plane).

A combination of visualization and attachment energy calculations (*E_att_*) could provide a more accurate prediction of the slip system compared to either individual method. Therefore, both visualization and *E_att_* calculations methods were employed for reliable identification of slip systems (Table 3). *E_att_* values obtained from COMPASS II force field method are presented in Table 3. The results of *E_att_* calculations using “Dreiding force field method” are provided in the Appendix A file. The presence and absence of slip planes in the molecular solids was experimentally confirmed by the nanoindentation study and the face indexation data previously reported by the authors [10].

#### 4.5.1. Crystallographic Features of RIV-MAL Co

The visualization and *E_att_* methods predicted the slip along (0 0 1) plane in the RIV-MAL Co crystal structure (Figure 7 and Table 3). The visualization method showed the (0 0 1) plan to be a slip plane with a flat-layered topology (Table 4). The nature of hydrogen bonding is one-dimensional (1D) across each adjacent layer of the slip planes, with the (0 0 1) plane exhibiting the largest inter-planer distance (d-spacing) of 6.2127 Å when determined from simulated XRPD data. In addition to intramolecular H-bonding between RIV molecules, intermolecular H-bonding between RIV and MAL molecules was also observed in the crystal structure of RIV-MAL Co.

The growth morphology model showed the (0 0 1) plane having the lowest *E_att_*, indicating that the plane along this direction may have the weakest intermolecular interactions, hence it can be a probable slip system. Using the face indexing analysis, the facet corresponding to (0 0 1) plane was identified as morphologically the largest crystal facet of RIV-MAL Co, with a relative surface contribution of 57.9% to the crystal habit [10]. These results provide experimental evidence that supports the findings of visualization and *E_att_* methods. Further, the “pop-ins” observed in the nanoindentation (*p**‒h*) loading curve of RIV-MAL Co crystal provided additional evidence for the presence of active slip plane system in the crystal structure (Figure 8).

Pop-ins (burst in) is a result of discontinuities (a sudden increase of displacement at same load) in the load-displacement (*p**‒h*) curve and indicates plastic deformation mediated by the active slip planes in organic molecular solids. When indention stress is applied perpendicular to the slip plane (i.e., facet corresponding to slip plane), pop-ins can be observed due to the increased plasticity presented by the slip planes that are encountered during the penetration of nanoindenter tip. Conversely, a smooth curve without pop-in is indicative of the absence of a slip-plane system in a given crystal [29].

#### 4.5.2. Crystallographic Features of RIV

Visualization of the RIV crystal structure displayed the presence of (0 1 1) and (1 0 2) as the primary and secondary slip planes, respectively (Figure 9). The plane (0 1 1) is termed as the “primary slip plane” as it has comparatively larger d-spacing (4.4850 Å) than the (1 0 2) plane (3.8752 Å), which is considered as the “secondary slip-plane” (Table 4). As shown in Figure 9, the surface topology of (0 1 1) is a flat layer, while that of (1 0 2) is corrugated or zigzag layers. Similar to RIV-MAL Co, the RIV crystal structure possesses intermolecular 1D H-bonding between RIV molecules.

On the *E_att_* calculations, the lower attachment energies (close to the least *E_att_*) were observed for the two planes, i.e., (0 1 1) with *E_att_* = −44.2 kcal/mol, and (0 0 1) with *E_att_* = −39.9 kcal/mol (Table 3). This indicates the possibility of two primary active slip planes. However, when (0 0 1) plane is visualized using Mercury software, it shows strong intermolecular interactions ruling out the possibility of (0 0 1) being a slip plane. Moreover, face indexing analysis has previously reported that facet corresponding to (0 1 1) plane as morphologically the largest crystal facet with a relative surface contribution of 34.8% to the crystal habit of RIV [12]. This experimental evidence supports the presence of (0 1 1) as the primary slip plane. The *E_att_* of (1 0 2) plane was observed to be ‒83.3 kcal/mol. The presence of active slip plane in the crystal structure of RIV was further supported by the observed pop-ins in the nanoindentation (*p‒h*) loading curve (Figure 8).

#### 4.5.3. Crystallographic Features of MAL

The visualization of the crystal structure of MAL revealed the presence of interlocked 3D hydrogen bonding between MAL molecules (resembling isotropic interactions) (Figure 10). The possibility of slip (glide) is not likely in the MAL crystal structure owing to the presence of strong intermolecular interactions across all three crystallographic axes (x, y, and z). The lowest attachment energy was observed for (1 0 0) plane in the MAL crystal structure (*E_att_* = −13.3 kcal/mol) (Table 3). The slip system prediction using attachment energy calculations assumes any plane with the least *E_att_* as a slip plane [47,48,50]. However, this assumption may provide erroneous predictions as it does not consider the role of the strong hydrogen bonding interactions commonly present in a crystal structure with 3D interlocked molecules. Some planes in the 3D interlocked structure would obviously have lower interactions energies than other planes, hence one may consider the least attachment energy plane as the “slip plane” based on the calculated *E_att_*. Therefore, prediction of slip planes in a 3D interlocked crystal structure only based on Eatt calculations can be erroneous. For 3D interlocked crystal structure, one should rely on either visualization method or other experimental techniques such as nanoindentation. The visualization method indicated the absence of a slip plane in the crystal structure of MAL. Additionally, the smooth (*p‒h*) loading curve (without pop-ins) verified the absence of slip planes in the MAL crystal structure (Figure 8).

### 4.6. Decoding the Basis of Bulk Deformation Behavior: Impact of Crystallographic and Supramolecular Features

#### 4.6.1. Impact of Crystallographic Features on Plasticity

Based on the compressibility plots and mean yield pressures, the order of plasticity (i.e. BA) for the three solids is MAL < RIV < RIV-MAL Co. Thus, the observed plastic deformation can be qualitatively written as “good” for RIV-MAL Co (Py of 83 MPa), “average” for RIV (Py of 133 MPa) and “poor” for MAL (Py of 323 MPa).

The greater plasticity of the cocrystal can be ascribed to the presence of the (0 0 1) active slip plane, which facilitates the facile slip in a direction perpendicular to the applied compaction pressure. Moreover, the higher inter-planar distance (d-spacing, 6.217 Å) in the slip plane of the cocrystal crystal structure as compared to the slip plane in RIV crystal structure (d-spacing, 4.4850 Å) could help in better slippage of the planes (higher plasticity) in the cocrystal system.

The crystal structure evaluation of RIV unveiled the presence of two slip planes, i.e., (0 0 1) and (1 0 2) and one-dimensional hydrogen bonding (Table 4). It was expected that RIV would have excellent plasticity owing to the presence of the multiple slip systems. However, RIV was less plastic than RIV-MAL Co. This behavior can be explained by the presence of corrugated or zigzag topology of the secondary slip plane (1 0 2).

The corrugated topology has an adverse impact on plastic deformation due to possible intertwisting (interlocking) of these planes during gliding over other nearest plane(s). The unfavorable impact of corrugated slip planes on plastic deformation of materials was also reported in crystal systems like paracetamol Form I, sulfathiazole Form IV and sucrose [49]. The presence of corrugated slip planes may facilitate a new hydrogen bond formation during slippage (gliding) process. On one hand, the intermolecular interaction is weaker due to separation of layers, while on the other hand, the intertwisting (interlocking) brings two layers close enough for formation of new hydrogen bonds. The likelihood of interlocking depends upon the degree of interlayer separations [51]. The inter-planar distance, i.e., d-spacing, can be a good indicator of interlayer separation between two planes [49]. A lower value of d-spacing indicates lesser interlayer separation and vice versa. In the case of RIV, a low value of d-spacing (3.8752 Å) for (1 0 2) plane signifies less interlayer separation and increased chances of interdigitation. The plastic deformation due to the primary slip plane (0 1 1) could be adversely affected by the resistance offered by the corrugated slip planes (10‒2). Hence, the overall plasticity of RIV is governed by the presence of primary (flat layers topology slip) and secondary (corrugated topology slip) planes. Thus, RIV is a unique crystal system wherein both flat layers and corrugated slip planes are observed in the same system, and the BA is governed by both slip planes. Moreover, the higher elastic recovery of RIV crystals (demonstrated by nanoindentation) could have reduced the available BA during bulk deformation.

Crystal structure evaluation of MAL was carried out to investigate the cause of its profoundly lower plasticity. MAL molecules form multiple hydrogen bonds across all three crystallographic axes forming a 3D network of hydrogen bonds (Figure 10). The lower plastic deformation (Py of 323 MPa) could be a result of the absence of slip plane (Table 4) and the rigid crystal structure of MAL. The 3D hydrogen bonding network of MAL offers a closer crystal packing which confers a dense crystalline structure and the same is evidenced by its higher crystal packing density (crystallographic true density is 1.621 g/cm^2^). Closer packing of the molecules along with strong intermolecular interaction (3D hydrogen bonding network) presented a rigid structure, which resists deformation under compaction pressure, resulting in the poor compressibility (poor plastic flow) of MAL. Therefore, MAL exhibited lower densification and higher yield pressure at a given pressure. The MAL possessed a lower plasticity and this material could be brittle in nature based on its high mean yield pressure (Py).

#### 4.6.2. Impact of Crystallographic Features on BS

The measured BS in terms of σ_0_ was found to be higher in MAL (σ_0_ = 2.75 MPa) than RIV (σ_0_ = 2.58 MPa), while the highest BS was observed for RIV-MAL Co (σ_0_ = 3.21 MPa). The crystal systems devoid of slip planes showed a direct relationship between true density and compactibility (BS) and the inverse relationship between true density and compressibility (BA). This relationship holds true for the series of crystal systems studied in our lab [19,25,26] and is largely applicable to other systems [20,52]. Closely packed crystal systems have a higher crystallographic density (true density) and exhibit higher intermolecular interactions as compared to the crystal systems with an open structure and lower true density. Therefore, higher true density enables greater intermolecular contacts during compaction, resulting in higher BS. At the same time, the systems having higher true density resist densification under the applied compaction pressure and hence exhibit higher yield pressure and poor compressibility (low BA).

MAL crystal structure is devoid of slip planes and contains higher intermolecular interactions across all three axes (3D interlocked) that give rise to its higher true density of 1.621 g/cm^3^. The reasonably good BS and poor compressibility (poor plasticity) of MAL were thus attributed to its higher true density. Many reported studies have considered only slip plane or plasticity (increasing BA) as key to govern tabletability and have neglected the role of BS. It is necessary to understand the role of BS in governing tabletability of molecular solids, and also considering “true density” as an important parameter, as it has a direct correlation with BS.

In crystal systems with active slip planes, BS is expected to be directly proportional to the strength of intermolecular interactions across the lowest energy slip plane in the corresponding crystal form [24,31]. In the crystal structure of RIV-MAL Co, MAL molecules form additional intermolecular hydrogen bonding interactions between the carbonyl (C=O) group of morpholinone in RIV (H-bond acceptor) and hydroxyl (‒OH) in MAL (H-bond donor) [10]. Thus, cocrystallization of RIV with MAL led to the increased extent of intermolecular interactions in the crystal structure of RIV-MAL Co along the lowest energy slip plane (0 0 1). Thus, the significantly higher BS (σ_0_ = 3.21 MPa) of the cocrystal over RIV (σ_0_ = 2.58 MPa) can be attributed to the increased strength of intermolecular interaction. A summary of the key findings of the present work is pictorially captured in Figure 11.

## 5. Conclusions

Cocrystallization led to altered crystallographic (supramolecular) features as compared to the parent components owing to the creation of a new multi-component crystal phase, which resulted to improved tabletability. This work assessed particle (crystal) and bulk level deformation behaviour of the molecular solids containing no active slip plane (i.e., MAL), a single active slip plane (i.e., RIV-MAL Co) and two slip planes of different surface topologies (RIV).

The BA in RIV was attributed to different surface topologies (flat and corrugated) of two active slip plane systems and higher elastic recovery of RIV crystals. The higher true density and the higher degree of intermolecular interactions due to the 3D interlocked structure offered reasonably good BS to MAL compacts. Concurrently, the strong intermolecular interactions resisted the densification under applied compaction pressure and hence resulted in a decrease in BA. The API and coformer displayed poor particle and bulk deformation; however, RIV-MAL compacts exhibited higher BA and greater BS. The cocrystal with highest BA and BS demonstrated significantly highest tabletability amongst the three samples. The increasing BA was attributed to the presence of flat-layered slip plane and the higher inter-planar distance, while the higher BS was ascribed to the increased degree of intermolecular interactions. During tabletability evaluation of molecular organic solids, plasticity signifies the role of the only BA. The present work stresses the necessity of understanding the role of BS as well.

The particle level deformation parameter H/E was found to inversely correlate with bulk level deformation parameter σ_0_, which has been commonly used to indicate BS. We suggest the utility of this correlation for estimation of bulk deformation behaviour based on the crystal deformation behaviour studied using nanoindentation experimentation. The predictability of this relationship needs to be further verified by studying more organic molecular solids.

## Figures and Tables

**Figure 1 pharmaceutics-12-00546-f001:**
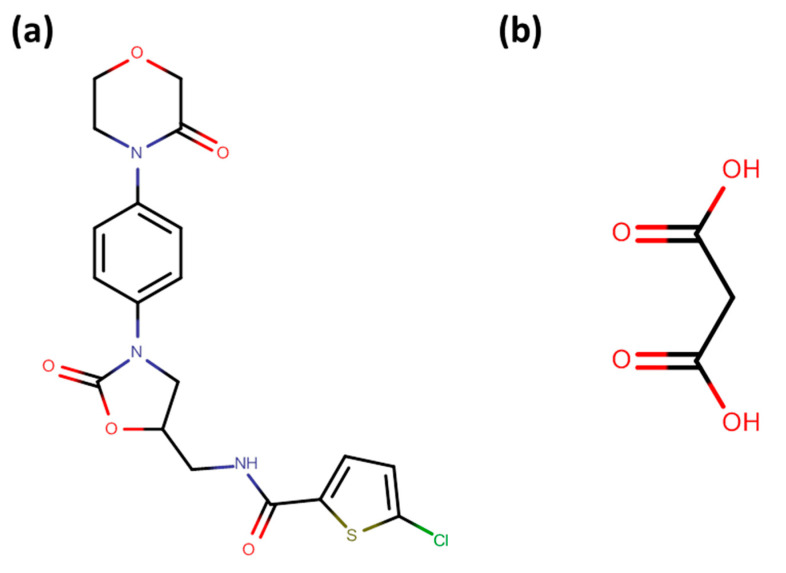
Molecular Structure of (**a**) RIV and (**b**) MAL.

**Figure 2 pharmaceutics-12-00546-f002:**
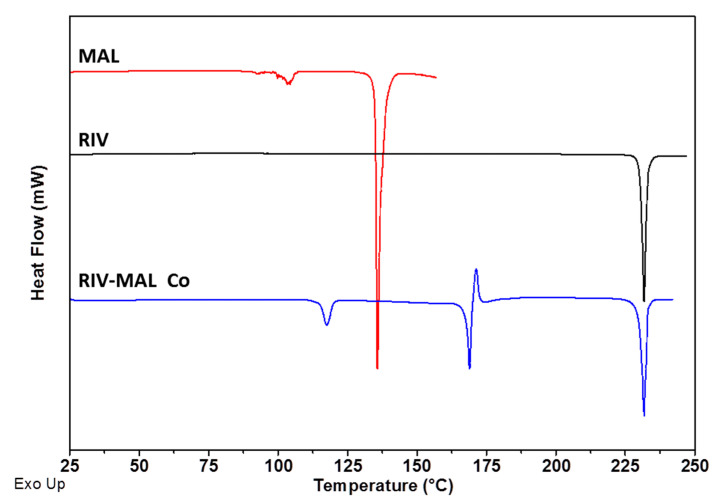
DSC heating curves overlay for RIV, MAL, and RIV-MAL Co Samples.

**Figure 3 pharmaceutics-12-00546-f003:**
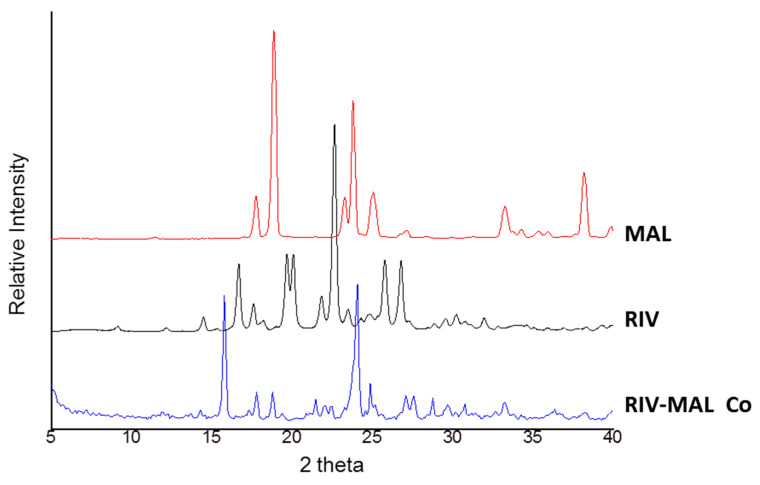
Overlay of XRPD diffractograms for RIV, MAL, and RIV-MAL Co Samples.

**Figure 4 pharmaceutics-12-00546-f004:**
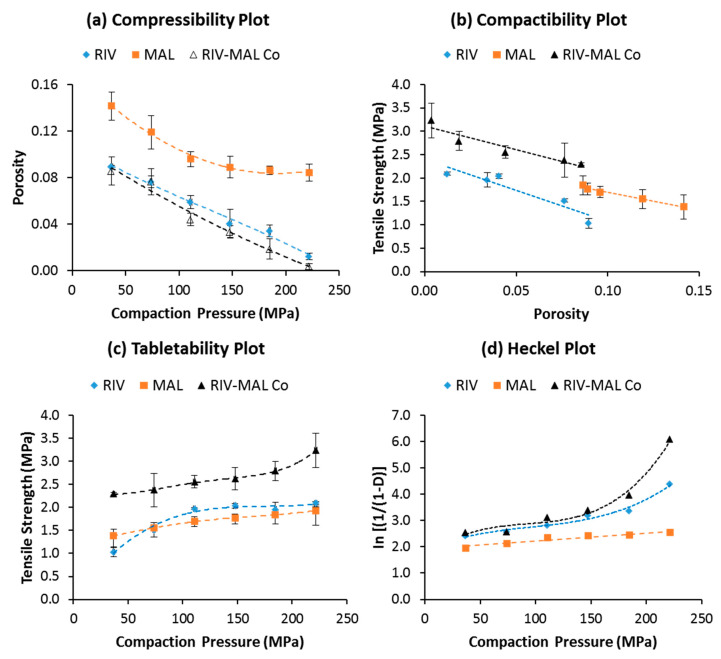
(**a**)–(**d**) Compressibility, tabletability, and compactibility, and Heckel plots for MAL, RIV, and RIV-MAL Co.

**Figure 5 pharmaceutics-12-00546-f005:**
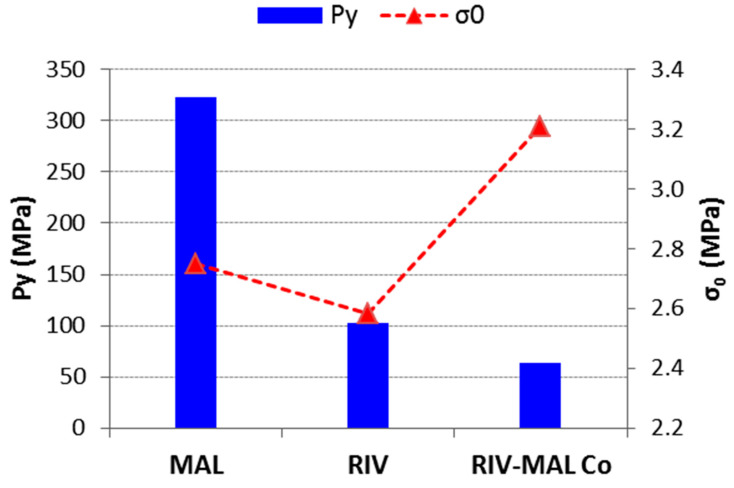
A plot of mean yield pressure (Py) and tensile strength at zero porosity (σ_0_) for RIV, MAL, and RIV-MAL Co.

**Figure 6 pharmaceutics-12-00546-f006:**
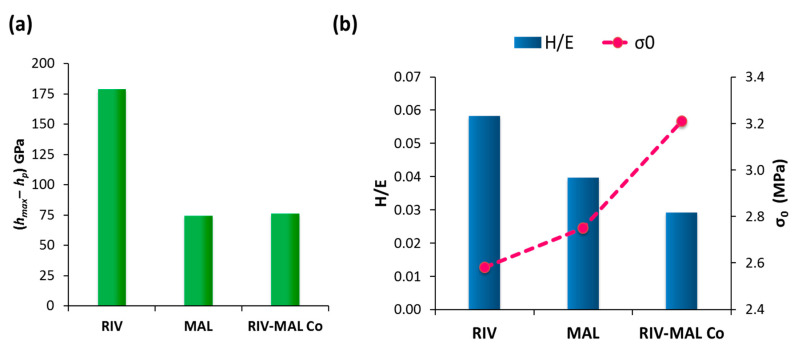
Nanoindentation parameters for crystal deformation representing (**a**) The (*h_max_**‒h_p_*) values depicting elastic recovery for crystals of RIV, MAL, and RIV-MAL Co; (**b**) Inverse correlation of H/E ratio (primary axis) and interparticulate bonding strength, σ_0_ (secondary axis).

**Figure 7 pharmaceutics-12-00546-f007:**
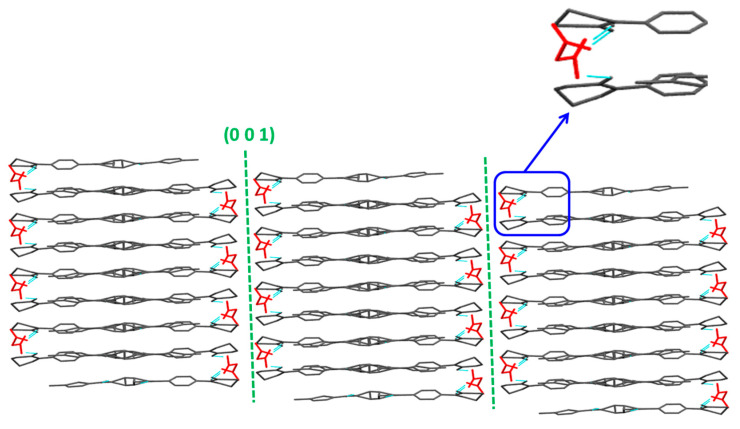
Crystal structure of RIV-MAL Co revealing the presence of flat layers topology slip plane along the (0 0 1) plane. MAL molecules are shown as “red”, RIV in “black” and hydrogen bonding interaction between RIV and MAL as “cyan”. The “insert” shows increased intermolecular interactions along the weakest crystallographic plane due to incorporation of MAL in the crystal lattice.

**Figure 8 pharmaceutics-12-00546-f008:**
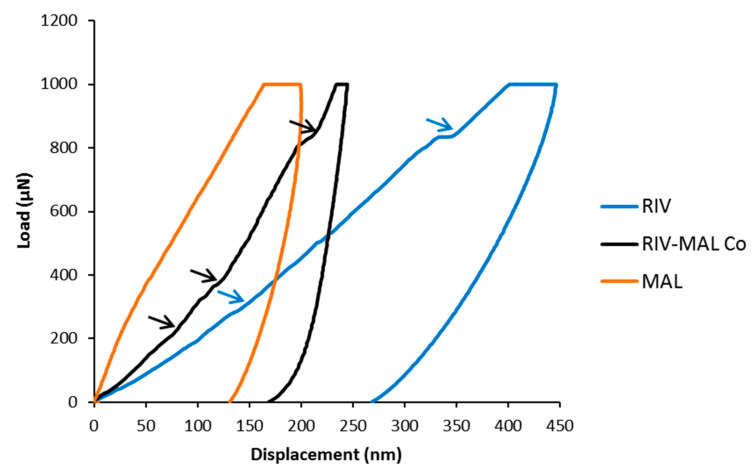
The load-displacement (*p**‒h*) curve for RIV, MAL, and RIV-MAL Co. The pop-ins observed in crystal samples of RIV and RIV-MAL Co are shown by pointing “blue” and “black”, arrows, respectively.

**Figure 9 pharmaceutics-12-00546-f009:**
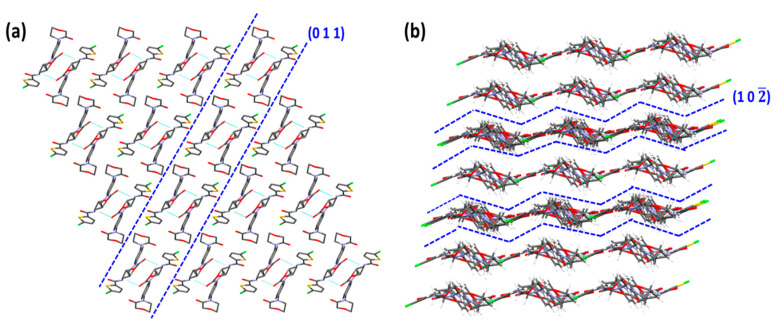
Crystal structure evaluation of RIV indicating the presence flat layers of RIV molecules with 1D hydrogen bonding along (0 1 1) plane (**a**), and corrugated layers along (1 0 2) plane (**b**).

**Figure 10 pharmaceutics-12-00546-f010:**
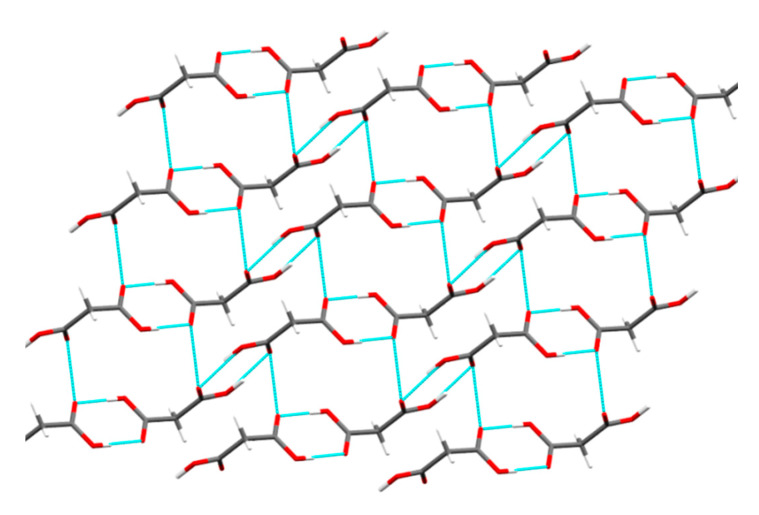
Crystal structure of MAL illustrating 3D hydrogen bonding network with strong intermolecular interactions.

**Figure 11 pharmaceutics-12-00546-f011:**
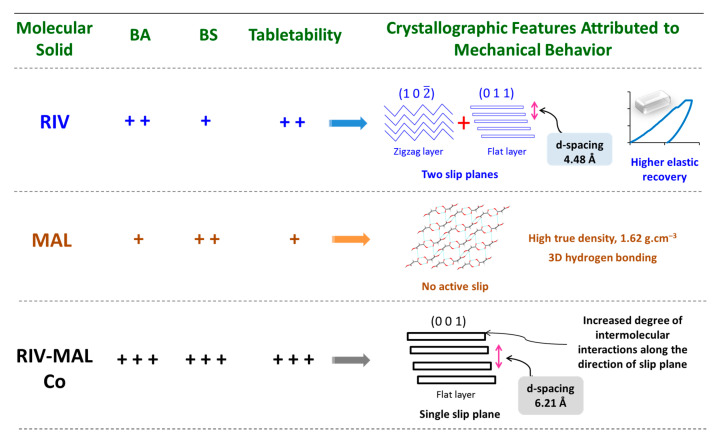
Pictorial depicting a summary of the key findings of present work. The observed values of interparticulate bonding strength (BS), bonding area (BA), and tabletability are expressed in “+” symbol. For the purpose of comparison, “+ + +” indicate “high”; “+” indicate “low”; and “+ +” represents “intermediate” value.

**Table 1 pharmaceutics-12-00546-t001:** Physical Characterization of MAL, RIV, and RIV-MAL Co.

Material	PSD (µm)			SSA (m^2^/g)	MC (% *w/w*)	True Density (g/cm^3^)	
	*D_10_*	*D_50_*	*D_90_*			Experimental	Crystallographic
MAL	8.3	14.2	37.8	0.78 (0.09)	0.264 (0.013)	1.628 (0.001)	1.621
RIV	7.1	10.2	31.5	0.85 (0.04)	0.235 (0.014)	1.536 (0.005)	1.554
RIV-MAL Co	7.8	11.9	35.2	0.83 (0.07)	0.242 (0.012)	1.534 (0.007)	1.548

Values in parentheses indicate standard deviation (SD), n = 3. PSD—Particle size distribution, SSA—Specific surface area, MC—Moisture content.

**Table 2 pharmaceutics-12-00546-t002:** Elastic modulus (E), mechanical hardness H, and 1/E values for crystal samples.

Sample	H (GPa) ^a^	E (GPa) ^a^	1/E
RIV	0.20 (0.02)	3.41 (0.24)	0.293
MAL	0.71 (0.08)	17.91 (2.35)	0.056
RIV-MAL Co	0.51 (0.04)	17.58 (0.42)	0.057

^a^ Average values are presented, while standard deviations are shown in parentheses (n = ≥10).

**Table 3 pharmaceutics-12-00546-t003:** Identification of slip plane using visualization, *E_att_* calculation and nanoindentation studies.

Materials	CCDC Code	Slip Planes Identification by		Nanoindentation (Is Slip System Present?)
		Visualization	Attachment Energy (hkl), *E_att_* in kcal/mol ^$^	
RIV-MAL Co	1854618	(0 0 1)	(0 0 1), −28.4	present
RIV	1854617	(0 1 1) (1 0 2)	(0 1 1), −44.2 (0 0 1), −39.9 (1 0 2), −83.3	present
MAL	1209218	absent	(1 0 0), −13.3	absent

^$^*E_att_* values reported here are from COMPASS II force field method.

**Table 4 pharmaceutics-12-00546-t004:** Comparative assessment of molecular and supramolecular attributes of the coformer, API and cocrystal.

Materials	H-Bonding Dimensionality	No. of Slip Planes	d-Spacing (Å) and Slip Plane	Surface Topology
RIV-MAL Co	1D	1	6.2127, (0 0 1) *	Flat layers
RIV	1D	2	4.4850, (0 1 1)	Flat layers
			3.8752, (1 0 2)	Corrugated/Zigzag layers
MAL	3D	0	-	Network of hydrogen bonds

* (0 0 1) plane was identified as (0 0 4) in the simulated PXRD scan of RIV.

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
