# Peer review of "The Role of Cocrystallization-Mediated Altered Crystallographic Properties on the Tabletability of Rivaroxaban and Malonic Acid"

_pharmaceutics, 2020, doi:10.3390/pharmaceutics12060546_

Round 1

Reviewer 1 Report

In the submitted manuscript, authors have investigated the potential effect of co-crystallization on the tableting properties of rivaroxaban. A comparison of the particle and bulk level mechanical properties of materials is of great importance.

The concept of the study is relevant, as authors themselves state that co-crystallization might serve as an effective tool for improvement in tableting properties of solid materials. However, co-crystallization of an API might give rise to concerns related to API´s biopharmaceutical properties such as solubility, dissolution rate, bioavailability, etc. In the case of rivaroxaban, it is not obvious from the introductory section that there is an actual need for improvement in rivaroxaban tableting properties. It is a relatively low dose drug (15 or 20 mg per tablet in marketed products), and in such instances, it is more reasonable to use directly compressible diluents to provide appropriate tableting properties. Therefore, from the pharmaceutical point of view, it is necessary to provide an explanation of why rivaroxaban and malonic acid were selected as models for the presented study. It might seem rather random selection, and readers would benefit greatly if authors could explain their rationale for the selection of these specific API and conformer. For a more general conclusion, I recommend the authors provide a framework for the selection of a suitable conformer for a given API based on API´s crystallographic and mechanical properties, if possible.

Authors have used an unusually long dwell time for compaction – 1 minute. There should be an explanation for the selection of such a long dwell time. In the pharmaceutical industry, if dwell times are needed during compaction they are in the range of seconds. In my opinion, since the holding times in nanoindentation studies were also just several seconds, it would be reasonable for the powder compaction to be performed also at reasonable dwell times.

Authors have presented compressibility, compactibility, and tabletability plots for API, conformer, and respective co-crystal. I recommend authors bear in mind recommended values for the solid fraction (15%) i.e. porosity (85%) of pharmaceutical solids. Of course, co-crystal would be mixed with other excipients but the obtained porosity is very low, even at low compaction pressures, and it might influence (sustain) the dissolution rate. Therefore, authors should comment on the relevance of the values obtained for the co-crystal, in terms of the potential effect on API dissolution rate.

I recommend authors to take into account brittle deformation for materials with high mean yield pressure (Py). Brittle deformation may also contribute to the BA.

Author Response

We are grateful to the reviewers for their valuable suggestions, which indeed helped us to improve the quality of the manuscript. Following are our pointwise responses to address the reviewers’ comments. The required changes have been made in the revised manuscript and can be seen using the ‘Track Changes’ function in the Microsoft word file.

REVIEWER: #1

1) In the submitted manuscript, authors have investigated the potential effect of co-crystallization on the tableting properties of rivaroxaban. A comparison of the particle and bulk level mechanical properties of materials is of great importance. The concept of the study is relevant, as authors themselves state that co-crystallization might serve as an effective tool for improvement in tableting properties of solid materials. However, co-crystallization of an API might give rise to concerns related to API´s biopharmaceutical properties such as solubility, dissolution rate, bioavailability, etc. In the case of rivaroxaban, it is not obvious from the introductory section that there is an actual need for improvement in rivaroxaban tableting properties. It is a relatively low dose drug (15 or 20 mg per tablet in marketed products), and in such instances, it is more reasonable to use directly compressible diluents to provide appropriate tableting properties.

Therefore, from the pharmaceutical point of view, it is necessary to provide an explanation of why rivaroxaban and malonic acid were selected as models for the presented study. It might seem rather random selection, and readers would benefit greatly if authors could explain their rationale for the selection of these specific API and conformer. For a more general conclusion, I recommend the authors provide a framework for the selection of a suitable conformer for a given API based on API´s crystallographic and mechanical properties, if possible.

Authors Response:

We appreciate the reviewer viewpoint on the providing the rationale for choosing the low dose API (rivaroxaban) as a model system because the poor compaction property associated with a low dose drug can be manipulated by use of directly compressible diluents.

Oral bioavailability of rivaroxaban (RIV) is solubility and dissolution rate-limited at 20 mg dose strength. Therefore, a number of cocrystals of RIV using different coformers were prepared and characterized to improve biopharmaceutical performance of the API. During the course of crystal structure evaluations, we observed interesting and unique crystallographic features in the crystal structures of RIV, MAL and RIV-MAL Co. The MAL crystal structure is devoid of active slip plane, while crystal structures of RIV-MAL Co and RIV have one and two slip plane respectively. Therefore, these three solids were selected as “model systems” to understand the role of these crystallographic feature/properties on particle and bulk level deformation behavior. The aforementioned rationale of selecting three molecular solids in the present study is mentioned in the manuscript (kindly refer to the last paragraph of the introduction, on page 6).  Thus, this study has implications in establishing a structure-property relationship.

2) Authors have used an unusually long dwell time for compaction – 1 minute. There should be an explanation for the selection of such a long dwell time. In the pharmaceutical industry, if dwell times are needed during compaction they are in the range of seconds. In my opinion, since the holding times in nanoindentation studies were also just several seconds, it would be reasonable for the powder compaction to be performed also at reasonable dwell times.

Authors’ Response: The authors agree that a dwell time of less than second, is required for tableting on a high-speed rotary press machine in the pharmaceutical industry. On rotary press machine, a tablet (compact) of sufficient tensile strength can be prepared because the powder bed receives an instantaneous or sudden compression force within milliseconds. As the rate of applied force remains very high, this makes tableting possible with the short dwell time without any defects. On the other hand, when the powder is compressed using a hydraulic press (as carried out in present work), the rate of transfer of applied force is low and gradual, therefore required significantly higher dwell time (from 30 to 120 sec). When an excellent plastic material like microcrystalline cellulose compressed on a manual hydraulic press with short dwell time (few seconds), compacts could not be formed. Since the purpose of the study was to compare the bulk deformation of three solids and its subsequent correlation with crystallographic properties, the higher dwell time used will impact not have the scientific outcomes of the present study. The previous researchers have also used a dwell time up to 2 min in preparing compacts for “out-of-die” bulk deformation analysis. 1-3

References:

  • Yadav JP et al. Correlating single crystal structure, nanomechanical, and bulk compaction behavior of febuxostat polymorphs. Pharmaceutics 2017, 14, 866−874.
  • Sun C and Grant DJW. Influence of crystal structure on the tableting properties of sulfamerazine polymorphs. Pharmaceutical Research, 2001, 18 (3), 2001 274-280.
  • Yadav JP, Bansal AK, and Jain S. Molecular understanding and implication of structural integrity in the deformation behavior of binary drug−drug eutectic systems. Pharmaceutics 2018, 15, 1917−1927

3) Authors have presented compressibility, compactibility, and tabletability plots for API, conformer, and respective co-crystal. I recommend authors bear in mind recommended values for the solid fraction (15%) i.e. porosity (85%) of pharmaceutical solids. Of course, co-crystal would be mixed with other excipients but the obtained porosity is very low, even at low compaction pressures, and it might influence (sustain) the dissolution rate. Therefore, authors should comment on the relevance of the values obtained for the co-crystal, in terms of the potential effect on API dissolution rate.

Authors’ Response:

The present work was carried out with neat API, coformer and the cocrystal to establish a structure-property relationship between them. We agree that the final formulation would include the additional excipients that would influence the overall dissolution profile. The same was also demonstrated in additional studies (unpublished work).

4) I recommend authors to take into account brittle deformation for materials with high mean yield pressure (Py). Brittle deformation may also contribute to the BA.

Authors’ Response: We appreciate the reviewer suggestion on consideration brittle deformation in the present study. In the three molecular solids studied, MAL showed high Py values indicated its low plasticity and it might undergo brittle deformation. This point has now been included in the revised version of the manuscript (please refer the last paragraph on page no. 22, and the first few lines on page 35).

In this case, the brittle deformation can contribute to the BA and thus to the overall tabletability. However, the compactibility plot shows that at similar porosity values (i.e. similar bonding area at a porosity ~ 0.1), the tensile strength of MAL compacts is lower than that of the RIV-MAL Co compacts. This suggesting that, for MAL the increase in BA may not be associated with a significant increase in the BS (bonding strength).

Reviewer 2 Report

Pharmaceutics, Article 821135

Kale et al.

This paper presents the results of a comprehensive study of the crystallographic and tabletability of rivaroxaban and malonic acid and the cocrystal of the two compounds. The study is systematic and detailed and relates the mechanical properties (at the particle and bulk level) of the three materials to their crystal features, including presence or absence of slip plane(s), slip plane topology, separation between adjacent crystallographic layers and intermolecular interactions.

The paper certainly provides sufficient novelty and is of sufficient interest to warrant publication.

Overall there are a significant number of lapses in English – minor grammar inaccuracies- which the authors should carefully review and correct before the manuscript is published.

I have corrected a number of these in the annotated version of the article which I have attached.

I have some other minor corrections and comments which should be addressed before publication:

Abstract:

H/E should be defined here.

Methods:

More detail should be provided on the true density measurement methodology.

Page 12: A reference should be provided to support the statement that: “Indentations with contact depths of 1 order of magnitude larger than local surface roughness are thought to be sufficiently deep to avoid a strong effect of roughness on the measured mechanical properties”.

Results and discussion:

Page 14: The authors state: “The DSC heating curve of MAL sample showed a broad endothermic event between 85.0° to 109.0 °C (related to transition point) followed by a sharp melting endotherm at 135.0 °C …..”.

What type of transition is represented by the “transition point”, i.e. what is the nature of the transition?

Page 15: As above the authors state: DSC analysis of the RIV-MAL Co sample showed a transition point between 110 to 122 °C, followed by the endothermic peak corresponding to cocrystal melting at 167.9 °C….”.

What type of transition is represented by the “transition point”, i.e. what is the nature of the transition?

Page 16: The authors should state how (what techniques were used) to examine the compacts for solid-state phase stability.

Page 19: The fitted lines to the linear regions of the Heckel plots should be shown.

Author Response

RESPONSE TO THE REVIEWERS COMMENTS:

We are grateful to the reviewers for their valuable suggestions, which indeed helped us to improve the quality of the manuscript. Following are our pointwise responses to address the reviewers’ comments. The required changes have been made in the revised manuscript and can be seen using the ‘Track Changes’ function in the Microsoft word file.

This paper presents the results of a comprehensive study of the crystallographic and tabletability of rivaroxaban and malonic acid and the cocrystal of the two compounds. The study is systematic and detailed and relates the mechanical properties (at the particle and bulk level) of the three materials to their crystal features, including presence or absence of slip plane(s), slip plane topology, separation between adjacent crystallographic layers and intermolecular interactions.

Comments:

1) The paper certainly provides sufficient novelty and is of sufficient interest to warrant publication. Overall there are a significant number of lapses in English – minor grammar inaccuracies- which the authors should carefully review and correct before the manuscript is published. I have corrected a number of these in the annotated version of the article which I have attached.

Authors’ Response: We appreciate the reviewer for the positive assessment of our work. We thanks the reviewer for correcting the grammatical inaccuracies. Now, the manuscript has been revised for all English corrections.

2) Abstract:  H/E should be defined here.

Authors’ Response: As suggested, H/E has now been defined in the revised manuscript.

3) Methods: More detail should be provided on the true density measurement methodology.

Authors’ Response: The methodological details of true density measurement are included on page no. 10‒11 in the revised version.

4) Page 12: A reference should be provided to support the statement that: “Indentations with contact depths of 1 order of magnitude larger than local surface roughness are thought to be sufficiently deep to avoid a strong effect of roughness on the measured mechanical properties”.

Authors’ Response: The reference, which can exactly support this sentence, is not available. Therefore, the manuscript is now appropriately revised with new text which read as‒ “The sufficient contact depths, large enough to local surface roughness were estimated to avoid strong effect of roughness on the measured mechanical properties”. Please refer page no. 13.

5) In Results and discussion: Page 14: The authors state: “The DSC heating curve of MAL sample showed a broad endothermic event between 85.0° to 109.0 °C (related to transition point) followed by a sharp melting endotherm at 135.0 °C …..”. What type of transition is represented by the “transition point”, i.e. what is the nature of the transition?

Authors’ Response: The nature of transition was ‘solid-solid phase transition’. This information is now included in the revised manuscript (Page 15, in the last paragraph) and appropriate reference1 has been cited. The cited reference, however, did not provide details on polymorphic form change at the transition point.

Reference:

  1. Caires FJ et al.,. Thermal behaviour of malonic acid, sodium malonate and its compounds with some bivalent transition metal ions. Acta 2010, 497, 35-40.

6) Page 15: As above the authors state: DSC analysis of the RIV-MAL Co sample showed a transition point between 110 to 122 °C, followed by the endothermic peak corresponding to cocrystal melting at 167.9 °C….”.What type of transition is represented by the “transition point”, i.e. what is the nature of the transition?

Authors’ Response: Solid-solid phase transition was observed between 110 to 122 °C. This has been discussed in our previously published article (Kale DP et al.,. Mol. Pharm. 2019, 16, 2980-2991). This information is now incorporated in the revised manuscript and the reference of the published work has been cited (page no. 16, in the last paragraph).

7) Page 16: The authors should state how (what techniques were used) to examine the compacts for solid-state phase stability.

Authors’ Response: We are thankful to the reviewer for suggesting this point as it would help to increase the readability of the manuscript. DSC analysis was used to examine the solid-state phase stability of the compact after compression. This information has been included in the revised manuscript (page no. 18, in the last paragraph).

8) Page 19: The fitted lines to the linear regions of the Heckel plots should be shown.

Authors’ Response: As advised, this information has been provided in the supplementary information file (page no. 2‒3, Figure No. S1‒S3).
